# Pathological Response and Immune Biomarker Assessment in Non-Small-Cell Lung Carcinoma Receiving Neoadjuvant Immune Checkpoint Inhibitors

**DOI:** 10.3390/cancers14112775

**Published:** 2022-06-02

**Authors:** Frank Rojas, Edwin Roger Parra, Ignacio Ivan Wistuba, Cara Haymaker, Luisa Maren Solis Soto

**Affiliations:** Translational Molecular Pathology Department, The University of Texas MD Anderson Cancer Center, Houston, TX 77030, USA; frrojas@mdanderson.org (F.R.); erparra@mdanderson.org (E.R.P.); iiwistuba@mdanderson.org (I.I.W.); chaymaker@mdanderson.org (C.H.)

**Keywords:** major pathological response, neoadjuvant immunotherapy, biomarkers

## Abstract

**Simple Summary:**

Recently, the U.S. Food and Drug Administration (FDA) approved neoadjuvant immunotherapy plus chemotherapy for the treatment of resectable non-small-cell lung carcinoma (NSCLC) due to the clinical benefits reported in several clinical trials. In these settings, the pathological assessment of the tumor bed to quantify a pathological response has been used as a surrogate method of clinical benefit to neoadjuvant therapy. In addition, several clinical trials are including the assessment of tissue-, blood-, or host-based biomarkers to predict therapy response and to monitor the response to neoadjuvant treatment. In this manuscript, we provide an overview of current recommendations for the evaluation of pathological response and describe potential biomarkers used in clinical trials of neoadjuvant immunotherapy in resectable NSCLC.

**Abstract:**

Lung cancer is the leading cause of cancer incidence and mortality worldwide. Adjuvant and neoadjuvant chemotherapy have been used in the perioperative setting of non-small-cell carcinoma (NSCLC); however, the five-year survival rate only improves by about 5%. Neoadjuvant treatment with immune checkpoint inhibitors (ICIs) has become significant due to improved survival in advanced NSCLC patients treated with immunotherapy agents. The assessment of pathology response has been proposed as a surrogate indicator of the benefits of neaodjuvant therapy. An outline of recommendations has been published by the International Association for the Study of Lung Cancer (IASLC) for the evaluation of pathologic response (PR). However, recent studies indicate that evaluations of immune-related changes are distinct in surgical resected samples from patients treated with immunotherapy. Several clinical trials of neoadjuvant immunotherapy in resectable NSCLC have included the study of biomarkers that can predict the response of therapy and monitor the response to treatment. In this review, we provide relevant information on the current recommendations of the assessment of pathological responses in surgical resected NSCLC tumors treated with neoadjuvant immunotherapy, and we describe current and potential biomarkers to predict the benefits of neoadjuvant immunotherapy in patients with resectable NSCLC.

## 1. Introduction

Lung cancer is the leading cause of cancer incidence and mortality worldwide. In recent years, the treatment of non-small-cell carcinoma (NSCLC) has changed dramatically, thanks to immunotherapy and the discovery of oncogenic driver alterations that led to the development of molecular targeted therapy. These two milestones have significantly increased the survival and quality of life of lung cancer patients [1,2]. Surgical resection is the standard of care for stage I and II NSCLC, and it is also considered a multimodality approach for stage IIIA disease [3,4,5,6]. Though adjuvant and neoadjuvant chemotherapy have been used in the perioperative setting, there are considerable side effects from some chemotherapy drugs such as platinum-based compounds, and the five-year survival rate only improves by about 5% [7,8]. Neoadjuvant molecular-targeted therapy can reduce the risk of recurrence; however, a complete pathological response (cPR) was not observed [9]. Neoadjuvant treatment with immune checkpoint inhibitors (ICI) for resectable NSCLC, on the other hand, is now of high interest due to higher response rates, improved survival benefits, and better tolerability [10,11].

Recently, the U.S. Food and Drug Administration (FDA) approved neoadjuvant nivolumab in combination with platinum-doublet chemotherapy for resectable NSCLC based on the results of the phase 3 Checkmate 816 clinical trial that showed an improved event-free survival and a higher cPR in patients treated with the combination, as compared to chemotherapy alone, in stage IB-IIIA NSCLC patients [12]. This phase 3 clinical trial was preceded by several clinical trials that have shown the feasibility and efficacy of several immunotherapeutic approaches for neoadjuvant therapy alone or in combination in patients with resectable NSCLC Stage IB, II, and III. These studies have shown a low toxicity and a high percentage of patients achieving a major pathological response (MPR) and cPR [12,13,14,15,16,17].

The rationale of using immunotherapy as a neoadjuvant treatment lies in the concept that the administration of an ICI while the primary tumor is still in the patient will result in a better systemic anti-tumor immune response. Preclinical in vivo studies in murine models of breast and lung cancer have shown that neoadjuvant immunotherapy works better than immunotherapy in the adjuvant setting, supporting the hypothesis that ICI therapy would be more efficient in driving the anti-tumor T cell response when the tumor mass contains a high antigen burden to be recognized by host T cells [13,14,15].

One of the major challenges to measuring the clinical benefit in clinical trials of neoadjuvant treatment in patients with resectable NSCLC is the long process associated with measuring overall survival, which is the gold-standard outcome measure for clinical trials [16]. Pathological response (PR) has been proposed as a surrogate indicator of benefit to neaodjuvant therapy in order to expedite assessment of the effectiveness of the agent being tested in a clinical trial setting [17]. Although the correlation of PR with overall survival is still being evaluated, many investigators have proposed a standardized scoring system that accounts for the percentage of viable malignant cells and other histological features to evaluate the quality of the immune response after treatment [17,18].

The histopathological changes, molecular mechanisms of lung cancer biology, and the immune pathways driving or suppressing the anti-tumor immune response are key aspects in identifying biomarkers that can help to better stratify patients for immunotherapy- and targeted therapy-based approaches [19,20]. Preliminary data generated by clinical trials for patients with resectable NSCLC treated with PD-1 blockade and/or CTLA4 inhibition have reported histologic features in the tumor bed that are associated with response to immunotherapy. They have assessed tissue-, blood-, and host-based biomarkers that can potentially be used both as predictors of the benefit of therapy and in monitoring cancer progression at different time points [21,22,23].

In this review, we provide relevant developments for the assessment of pathological responses in surgical resected NSCLC tumors treated with neoadjuvant immunotherapy, and we describe the biomarkers that have been used to predict the benefit of neoadjuvant immunotherapy in patients with resectable NSCLC, as well as potential biomarkers that can be used to better stratify these patients for immunotherapy.

## 2. Pathological Response

The pathological response (PR) assessment consists of a histopathological evaluation of the extension of viable tumor cells in the tumor bed of surgically resected primary tumors after neoadjuvant treatment [17,24]. This assessment has been widely used as a surrogate method to indicate response to therapy and to expedite the assessment of clinical trials in several types of tumors from patients who underwent surgical resection after neoadjuvant chemotherapy, including breast carcinoma, melanoma, and locally advanced NSCLC [25,26,27]. In patients with NSCLC treated with neoadjuvant chemotherapy, tumors with a PR value of 10% or less are considered to have achieved MPR, and tumors with an absence of viable tumor cells are considered to have a pCR [28]. MPR has been associated with long-term overall survival (OS) in retrospective studies of several clinical trials of neoadjuvant chemotherapy [17,29,30,31].

The pathological features of NSCLC tumor samples from patients treated with neoadjuvant chemotherapy have been studied by several investigators [17,29,30,31]. A detailed outline of recommendations from The International Association for the Study of Lung Cancer (IASLC) has been released to standardize assessment and allow for comparison among new therapeutic options that are tested in clinical trials, such as tyrosine kinase inhibitors or immunotherapy with ICIs [28]. These recommendations provide strategies for pathological specimen processing for the proper recognition of the tumor bed, including correlation with a computed tomography scan and surgeon marks, and for proper sampling for the microscopic assessment of pathological response. The outline also includes recommendations for a histological evaluation to delineate the tumor bed and surrounding non-neoplastic lung, and for the determination of PR by histological quantification of the viable tumor, necrosis and stroma, along with the correlation of microscopic findings with gross examination and mapped gross photographs [21,28,32]. For routine cases, these guidelines do not recommend either using IHC as an aid in the recognition of tumor cells from other non-malignant cells or using computational tools [28]. Similarly, for clinical reporting, the histological features of fibrosis and inflammation are not required, although these features have been described in several studies using lung cancer specimens after neoadjuvant chemotherapy, and they have been acknowledged as components of tumor-intrinsic changes related to immunological mechanisms [28,32].

PR has also been evaluated in metastatic lymph nodes resected after neoadjuvant chemotherapy, and studies have shown that patients with MPR-positive lymph nodes have a better survival than those with MPR-negative lymph nodes; however, there is still no consensus on cut-offs to define MPR in lymph nodes [33,34]. Of note, Pater et al. [33] showed that, among patients with resectable NSCLC treated with neoadjuvant chemotherapy that did not achieve MPR in primary tumors, patients with MPR-positive lymph nodes, defined as a percentage of the viable tumor <70%, have better outcomes than patients with MPR-negative lymph nodes, suggesting that this information may help to better stratify patients for prognosis. Nevertheless, the clinical impact of evaluating MPR in lymph nodes, as well as its diagnostic reproducibility among pathologists, is still under investigation [28].

Of note, several histological changes in the tumor microenvironment of tumor specimens, such as prominent elastic fibers, from patients that have previously been exposed to chemotherapy or chemoradiotherapy, can also be found in therapy-naïve adenocarcinoma, as well as in other pathological processes such as pleuroparenchymal fibroelastosis and bone marrow and lung transplantation [28]. Histological changes, such as inflammation of the blood vessels, medial fibrotic thickening, and cytological atypia, have also been observed in tumors after chemotherapy and radiotherapy [28]. However, coagulative necrosis, foam cell infiltration, and inflammatory infiltrates can also be found in resected tumors from both groups of patients, those who received chemotreatment and those who were treatment-naïve [24].

It has also been suggested that some chemotherapy agents, including cisplatin combined with gemcitabine or docetaxel and carboplatin combined with paclitaxel or pemetrexed, activate the immune response of NSCLC patients, since PD-L1 and specific T-cells are higher in NSCLC resected tumors that received neoadjuvant chemotherapy compared to chemo-naïve tumors [32]. Likewise, similar features can also be found in resected specimens after immunotherapy. These types of specimens show, in addition, a higher number of tumors infiltrating lymphocytes and tertiary lymphoid structures [35].

### 2.1. Pathological Response Evaluation in Tumors after Neoadjuvant Immunotherapy

Limited studies have been performed in tumor samples from surgical resected NSCLC treated with immunotherapy. The evaluation of these specimens has the potential to provide information not only about the utility of the percentage of the viable tumor as a surrogate endpoint of clinical benefit, but also about the type and quality of the immune response in the tumor bed. Therefore, this information may allow for the establishment of a better understanding of the mechanisms associated with immune response and resistance to therapy [35,36].

Similar to neoadjuvant chemotherapy-treated NSCLC, for neoadjuvant immunotherapy, MPR and cPR have been used as surrogate predictors of survival in clinical trials with neoadjuvant immunotherapy alone or in combination with chemotherapy. MPR varied from 14% in the PRINCEPS trial (ClinicalTrials.gov Identifier: NCT02994576; stage I-IIIA NSCLC treated with neoadjuvant Atezolizumab) to 50% in the NEOSTAR trial (ClinicalTrials.gov Identifier: NCT03158129; stage I-IIIA NSCLC treated with nivolumab and Ipilimumab) [37,38], as shown in Table 1. In clinical trials with neoadjuvant immunotherapy in combination with chemotherapy, MPR was achieved in 36,9%, 57%, and 83% in the Checkmate 816 (ClinicalTrials.gov Identifier: NCT02998528; stage IB-IIIA NSCLC treated with nivolumab + chemotherapy), MAC (ClinicalTrials.gov Identifier: NCT02716038; stage IB-IIIA NSCLC treated with atezolizumab + chemotherapy), and NADIM (ClinicalTrials.gov Identifier: NCT03081689; stage IIIA NSCLC treated with nivolumab + chemotherapy) trials, respectively [25,39,40].

A detailed histopathological assessment of the tumor bed was performed by Cottrell et al. [35] in 20 resected primary tumor specimens from patients with NSCLC treated with neoadjuvant immunotherapy. In this study, a proposal for an immune-related pathologic response (irPR) system was developed that defines the tumor bed as the sum of the residual viable tumor, necrosis, and regression bed. The regression bed was defined by “proliferative fibrosis with neovascularization and evidence of immune activation and cell death” (Figure 1). In this assessment, the intratumoral stroma with no histological features of regression is counted as residual viable tumor, and features of cell death, immune activation, and tissue-repair phenomena were recorded in each sample [22,41,42]. Elements from each category—specifically, immune activation features such as higher tumor immune infiltration (TIL) score, tertiary lymphoid structures (TLS) (lymphoid aggregates with a germinal center and high endothelial venules), plasma cell infiltrates (≥100 plasma cells /high power field (HPF) in at least two HPF), cell death features such as the presence of cholesterol clefts and foamy macrophages, and tissue-repair features such as proliferative fibrosis and neovascularization [35] (see Figure 2)—were extensively observed in tumors from responder patients compared to non-responders.

The assessment of irPR using these criteria has been evaluated by other investigators in samples obtained from clinical trials of neoadjuvant immunotherapy. Ling et al. [36] characterized the histopathological features of the primary tumors and lymph nodes of 31 surgically resected lung squamous cell carcinoma after neoadjuvant treatment with anti-PD1 from patients of a phase Ib study of neoadjuvant anti-PD-1 (sintilimab) therapy (Registration Number: ChiCTR-OIC-17013726). In these samples, the authors observed similar histological features in the regression bed described by Cottrell et al. [35]. In all samples, including non-responders, Ling et al. [36] also classified the tumors in three immune phenotypes based on the presence of immune infiltration in the tumor area: immune-activated, i.e., a viable tumor area with immune infiltration in both tumor nests and surrounding tumor stroma; immune-excluded, i.e., viable tumors with immune infiltration only in the surrounding stroma; and immune-desert, i.e., viable tumors with an absence of immune infiltration (an example of these histological features is shown in Figure 3). Immunotherapy-treated samples were heterogeneous in these phenotypes; however, all samples that exhibit MPR showed an extensive immune-activated phenotype [35,36]. Interestingly, in these samples, in situ squamous cell carcinoma was observed in the tumor bed, which indicates that histological changes should be assessed considering tumor-specific biology [36]. The histological features of immune activation, cell death, and tissue repair, as defined by Cottrell et al. [35], were also observed across different tumor types treated with neoadjuvant immunotherapy, including NSCLC, cervical carcinoma, and melanoma, among others, suggesting that a universal approach that can aid in assessing these features across all types of tumors treated with immunotherapy, including primary tumors and metastasis, would allow for a better evaluation of treatment effect and cross comparison to elucidate possible mechanisms of resistance and response to therapy [18]. Different from chemotherapy-treated tumors, on immunotherapy-treated tumor samples, necrosis was not commonly seen, and it has been suggested that its presence may not be related to immunotherapy response. Further, a distinction between tumor-intrinsic necrosis and therapy effect may not be distinct in most cases [35,36].

The evaluation of tumors from patients that have been treated with neoadjuvant immunotherapy poses several challenges to clinicians assessing efficacy to immunotherapy, mainly because, after immunotherapy, tumors may show a temporary increase in tumor burden as assessed by imaging, and pathological assessments have shown that the increase in the size of these tumors may be due to the presence of immune cell infiltration, a phenomenon that is referred to as pseudo-progression [41,42,43]. This is significant due to the fact that, in several clinical trials of neoadjuvant-treated NSCLC, the lesion size evaluated by computed tomography (CT) scan differed from that with microscopic assessment [35,36]. Likewise, another phenomenon of pseudo-regression has been described in the lymph nodes of NSCLC patients treated with neoadjuvant ICIs: The nodal immune flare (NIF), which consists of abnormal lymph nodes, identified by an increased uptake of 18Fluorodeoxiglucose positron emission tomography/computer tomography, with an absence of tumor cells by microscopy assessment. Instead, these abnormal lymph nodes displayed an increase in nodal size and the presence of de novo non-caseating granulomas [14]. Cascone et al. [38] analyzed 72 patients who were treated with neoadjuvant ICIs in the randomized NEOSTAR trial (ClinicalTrials.gov Identifier: NCT03158129) and compared them with a subset of patients with abnormal lymph nodes found on upon imaging after neoadjuvant chemotherapy (ICON, ImmunogenomiC prOfiling in NSCLC cohort); this phenomenon was observed in 16% (7/44) of patients treated with ICIs and in 0% (0/28) of patients after neoadjuvant chemotherapy in ICON cohort patients. Worth mentioning is that 7% of the patients treated with ICI had nodes with de novo non-caseating granulomas on pathology analysis and were not radiologically abnormal. In this study, NIF was associated with a fecal abundance of intestinal flora of the genera belonging to Actinobacteria and Coriobacteriaceae [38]. The NIF phenomenon has not been associated with tumor responses or toxicity related to treatment [14].

### 2.2. Computational Pathology for the Assessment of MPR

The use of artificial intelligence (AI) and advanced computational techniques such as machine learning (ML) and deep learning may allow for obtaining additional information regarding the tumor contexture for the pathological assessment of these types of tumor samples. These techniques can also facilitate a more accurate and reproducible whole-slide image classification and tissue segmentation [39,44]. The capabilities of machine learning have recently expanded extensively due to the development of deep learning and convolutional neural networks (CNNs) [40]. One of the main promises of these computational tools is in their application as a clinical decision support for diagnosis. Artificial intelligence has been used in several pathology tasks, such as the identification of tumor cells [45], mitotic counts [46], immunohistochemistry scoring [47], and the spatial relationship of the tumor microenvironment [39,48]. It has also shown improved sensitivity in the lymph node metastases assessment [49].

Preliminary data of a study performed in 127 surgical resected specimens from neoadjuvant-treated NSCLC patients from a LCMC3 trial showed that the application of a machine learning-based approach to evaluate PR is feasible. Dacic et al. [50] quantified and measured the tumor bed area and identified residual viable tumor cells showing a strong correlation between the AI tool and manual MPR assessment, although with different percentage ranges, probably due to differences in methodology. However, no sensitive or positive predictive value has been presented [50].

Some things to take into consideration with the use of such advanced approaches include a significant investment in IT infrastructure, network limitations if the data are stored remotely or if the image processing is performed remotely, and the availability of reliable and variable training data [51,52,53]. Further studies that can address quantification and spatial information of the diverse cellular and architectural elements may be achieved using computational tools integrated with biomarker analyses of different immune cell types involved in the immune response changes of neoadjuvant-treated tumors.

## 3. Biomarkers for Potential Use as Predictors of Neoadjuvant Immunotherapy Efficacy

Several biomarkers have been studied in clinical trials with neoadjuvant immunotherapy for patients with resectable NSCLC. A summary of current and emerging biomarkers that are being studied in immune-oncology clinical trials is illustrated in Figure 4. A summary of assays that are used for immune-oncology related biomarkers is presented in Table 2.

### 3.1. PD-L1 Expression in Tumor and Stromal Cells

Programmed cell death-ligand 1 (also known as B7-H1 or CD274) is a transmembrane protein that is expressed in T cells, B cells, macrophages, dendritic cells, and tumor cells [13,54]. PD-L1 is the ligand of the programmed cell death 1 (PD-1; CD279) protein, which is expressed in T cells after chronic antigen stimulation. Engagement of the PD-L1/PD1 axis results in decreased T cell receptor (TCR) signaling and, subsequently, reduced activation, proliferation, cytokine secretion, and survival. Based on this mechanism, PD-L1 has been evaluated as a predictive biomarker for sensitivity to immune checkpoint blockade strategies targeting this axis [55]. PD-L1 expression assessed by immunohistochemistry in tumor tissue has been approved by the FDA as a standard biomarker for ICIs in NSCLC patients as a companion or complementary diagnostic test for different PD-1/PD-L1 inhibitor drugs [56].

In the neoadjuvant settings, PD-L1 expression has been investigated as a predictive biomarker in several clinical trials with disparate results [12,21,38,57,58,59]. In the phase 3 Checkmate186 trial, a considerably higher benefit was observed in patients treated with nivolumab and chemotherapy in tumors with ≥1% PD-L1 expression in tumor cells compared with tumors with <1% expression of PD-L1 [12]. Similarly, in the NEOSTAR trial, an analysis of PD-L1 assessed in tumor cells by IHC showed that, in pretreatment biopsies, PD-L1 was higher in patients with MPR and patients with radiographic responses compared to patients with no MPR and no radiographic response, respectively. No differences were observed in PD-L1 expression in resected tumors by MPR or radiographic response or in PD-L1 expression between pre-treatment biopsies and surgical resected tumors [38].

Other correlative studies from clinical trials of neoadjuvant immunotherapy in patients with resectable NSCLC did not show an association of PD-L1 expression in tumor cells with a clinical benefit of treatment [25,39,60,61,62]. In the LCMC3 trial, tumor regression and MPR were observed in patients with resectable NSCLC after neoadjuvant treatment with atezolizumab, regardless of the pretreatment status of PD-L1 expression [58,63]. Forde et al. did not find an association between pretreatment PD-L1 expression and MPR assessment in a cohort of 21 patients treated with neoadjuvant nivolumab [21]. In another study of neoadjuvant durvalumab alone or combined with stereotactic body radiation therapy, MPR was achieved independently of PD-L1 tumor status after adjusting for PD-L1 baseline expression assessed by IHC, and no significant changes in PD-L1 expression were observed when comparing pre-treatment and surgical resection tumor specimens in both trial groups and between patients with and without MPR [57]. Shu et al. also found that MPR was achieved in patients who received atezolizumab plus chemotherapy, regardless of their PD-L1 tumor expression [64]. Finally, Gao et al. [59] found that PD-L1 expression in the stromal cells of pretreatment samples correlated with pathological response in a cohort of patients that received neoadjuvant sintilumab, and no correlation of PD-L1 expression was found between pre-treatment and surgical resected specimens. The wide variability in the results among several small-scale studies might be related to different variables such as tumor histology, genomic features, tissue availability, neoadjuvant immunotherapy scheme, and tissue analysis for MPR [13].

### 3.2. Tumor Mutation Burden (TMB)

TMB, also known as tumor mutation load, is the total number of somatic missense mutations per coding area in the tumor genome. Overall, TMB correlates with the amount of predicted neoantigen load in a tumor. A higher amount of neoantigens contributes to a more effective anti-tumor T cell response, which is consistent with an immune inflamed phenotype observed in tumors with high TMB [65]. Lung tumors have higher frequencies of high TMB compared to other tumors, which is probably related to the mutagenic effects of tobacco smoking [60,66]. High TMB has been shown to be associated with high PD-L1, CTLA4, and CD8+ T cell infiltrates [61], as well as clinical benefit in advanced NSCLC patients that received neoadjuvant treatment with ICIs [62].

TMB is usually measured by whole exome sequencing (WES) or targeted next generation sequencing technology in tumor tissue. In 2020, the FDA approved the FoundationOneCDX assay (Foundation Medicine, Inc, Cambridge, MA, USA), a next generation sequencing test, as a companion diagnostic for pembrolizumab treatment in unresectable or metastatic TMB-high (TMB ≥ 10 mutations/megabase (mut/Mb)) solid tumors, including NSCLC [67]. TMB can also be measured in blood-based assays in circulating tumor DNA; however, its clinical utility as a predictive biomarker is still under development [68,69]. In addition to TMB, whole exome sequencing of tumor and matched normal DNA and RNA-seq whole transcriptome analysis, as well as computational bioinformatics, can also be used to predict neoepitopes that can induce an effective anti-tumor immune response, and to develop novel strategies for cancer immunotherapy [70,71].

TMB has been investigated in several clinical trials of neodjuvant immunotherapy in resectable NSCLC. In the phase 3 Checkmate 186 TRIAL, a greater benefit of Nivolumab plus chemotherapy was observed in patients with higher TMB compared to patients with low TMB (cutoff: 12.3 Mut/Mb by NGS TSO500 assay, which corresponds to 10Mut/Mb per the FoundationOne assay) [12]. Similarly, in the phase 2 pilot study ClinicalTrials.gov Identifier: NCT02259621, Forde et al. showed that, in a small set of 11 pretreatment samples, TMB was strongly associated with pathological response [21,63]. In this study, WES and the patient’s major histocompatibility complex class I haplotype was used to computationally predict mutation-associated antigen burden, and this correlated with pathological response [21]. On the other hand, in patients from the phase 2 LCMC3 trial, TMB was not associated with pathological response [12,63].

### 3.3. Oncogenic Driver Alterations

It is well known that oncogenic gene alterations play an important role in the initiation and development of NSCLC. Several of them are considered actionable, and targeted therapy of molecular subtypes, such as tumors with an *EGFR* mutation and *ALK* rearrangements, have become the standard of therapy for advanced NSCLC, and targeted therapies for other driver mutations such as *KRAS* are being evaluated in clinical trials [72,73].

There is limited data on the role of oncogenic driver alterations as a predictor of the benefit of neoadjuvant immunotherapy, since, in most clinical trials of neoadjuvant immunotherapy of resectable NSCLC, patients with tumors with driver alterations have been excluded based on the fact that patients with advanced NSCLC with actionable driver alterations have a lower or lack of response to an immune checkpoint blockade [74,75].

Preliminary results of the LCMC3 clinical trial showed that MPR was observed in zero of the 12 patients with either *EGFR* mutations (*n* = 7) or *ALK* fusions (*n* = 5) that underwent surgery. In this trial, MPR was observed in 11% of patients with *STK11/LKB1* mutations (1/9) and 22% (2/9) of patients with a *KEAP1* mutation, which was slightly lower than wild-type tumors (STK11/LKB1 wild type, MPR 17/71, 24%; KEAP1 wild type, MPR 16/71, 23%) [76]. In the pilot phase 2 trial, NCT02716038, of neoadjuvant atezolizumab and chemotherapy in resectable NSCLC, ten patients had known oncogenic drive alterations (*STK11*, 2; *KRAS*, 2; *KRAS/STK11*, 1; *EGFR,* 4; and *HER2*, 1). No patients with an *STK11* mutation achieved a partial response by RECIST criteria, and, in two of them, the surgical resection showed no MPR, and the third tumor was unresectable. In patients with *KRAS* mutations, one was unresectable, and one tumor achieved complete pathological response. In patients with the *EGFR* mutation, two of them, harboring exon 19 deletion and exon 20 insertion, did not achieve MPR, but two of them, harboring L858R and L858R/S7618I mutations, had a complete pathological response [64].

In the phase 2 trial, the ClinicalTrials.gov Identifier: NCT02259621 clinical trial of neoadjuvant Nivolumab or Nivolumab plus ipilimumab in resectable NSCLC, tumors with the *STK11* mutation were identified in six of the nine patients (66%). Three of them had progression, two of them harbored *KRAS* and *KEAP1* co-mutations and had disease progression precluding surgery, and one of them had a co-mutation with BRAF and *TP53* [77].

Currently, two clinical trials of neoadjuvant durvalumab, the AEGEAN trial (ClinicalTrials.gov Identifier: NCT03800134) and NeoCOAST trial (ClinicalTrials.gov Identifier: NCT03794544), are recruiting patients with no exclusion criteria for *EGFR/ALK* alterations. These trials will provide more insights into the efficacy of neoadjuvant immunotherapy in patients with oncogenic driver alterations.

A better understanding of the relations between tumor landscape and tumor immune response is vitally important to design studies to determine the benefit of neoadjuvant immunotherapy in resectable NSCLC. Specific genomic alterations can influence the tumor-immune landscape and have an impact on the clinical response to ICI. At the same time, the immune contexture of tumors may also influence the tumor mutational landscape in different ways, such as eliminating clones with strong antigenic neopeptides, influencing genomic diversity, and lung cancer evolution [78].

Given the recognized heterogeneity of lung cancer, a new model for lung cancer stratification based on the co-occurring genomic alterations of key genes has been proposed to define lung adenocarcinoma heterogeneity, which can aid in defining groups of tumors of similar immune microenvironments and potential specific therapeutic targets. For example, *KRAS/TP53* co-mutated tumors are associated with an inflamed immune microenvironment and increased tumor PD-L1 expression, whereas *KRAS/STK11* co-mutated tumors are characterized as being cold, with a lack of T cell inflammation and lower PD-L1 expression irrespective of TMB. In addition, the *STK11* mutation has emerged as a possible biomarker predictor of resistance to PD-1 and PD-L1 inhibition [78,79].

Further clinical trials that include oncogenic driver alterations will help to develop strategies for the categorization of patients with resectable NSCLC for immunotherapy treatment.

### 3.4. Tumor-Associated Immune Cells

Tumor-associated immune cells (TAIC) consist of multiple cell types that are part of the tumor microenvironment, including tumor-infiltrating lymphocytes (T cells, B cells), macrophages, natural killer (NK) cells, and dendritic cells, among others [80]. TILs in quantity and composition can serve as a predictive biomarker of the response to therapy and prognosis [81]. TILs can be assessed in tumor tissue using H&E slides by assessing the percentages of stromal TILs, intra-tumoral TILs, and TILs in the central tumor and invasive margin, following the guidelines of the International Immuno-oncology Biomarker Working Group. However, these guidelines have not been tailored to neoadjuvant immunotherapy-treated surgical samples [82,83].

Conventional immunohistochemistry and multiplex immunofluorescence and other high-plex platforms have been used in different studies to assess different immune cell subsets. This area is in constant development, and current technologies can allow for the quantification of specific immune cell types and spatial analysis in limited tissue samples [84,85,86]. In addition, techniques such as flow cytometry and mass cytometry can be used to phenotypically characterize immune cell subsets and functional states from single cell suspensions generated from tumor tissue.

In the context of neoadjuvant immunotherapy, there are limited studies that explore TILs and immune cell subsets. An initial exploratory analysis from the NEOSTAR trial found an increase in total CD3+ TILs as well as CD3+CD8+ TILs in resected tumors after neoadjuvant treatment with nivolumab plus ipilimumab in both MPR and non-MPR cases [38]). In addition, TILs from surgical resections following treatment with nivolumab plus ipilimumab had a higher percentage of tissue-resident memory-like CD8+ and CD4 T cells as well as CD4+ CD28+ CD27− T cells, as compared to nivolumab alone [38]. Early results from the LCMC3 trial showed a significant expansion of antigen-presenting cells, such as dendritic cells and B cell subsets, as well as a higher frequency of CD3+ CD27+ CD45RO+ T cell subsets, in the resected lymph nodes of patients with tumors that achieved MPR following neoadjuvant therapy with atezolizumab [13]. In the NADIM trial, multiplex immunofluorescence staining revealed that the majority of T cells were found in the stromal compartment as compared to the tumoral compartment, suggesting an immune-excluded phenotype. Interestingly, though a reduction in total CD3+ TIL was observed from diagnosis to the post-treatment surgical resections, an increase in CD3+ CD45RO+ T cells (effector/effector memory) was observed within the tumor compartment post-treatment. Within the stromal compartment, a highly significant reduction in the number of CD3+ T cells was observed post-chemotherapy. A stratification of cases based upon pathologic response at time of surgery showed that the presence of CD3+CD45RO+ TIL as well as CD3+ CD8+ CD45RO+ TIL correlated with improved patient outcomes, suggesting an association between the presence of effector/effector memory T cells and pathologic response [87].

### 3.5. The T Cell Receptor Repertoire

The T cell receptor (TCR) is a unique protein complex that recognizes antigens, including tumor neoantigens, that are bound to the major histocompatibility complex (MHC) as peptides molecules [88]. TCR repertoire features include density, diversity, and clonality, all of which have been evaluated as markers of the clinical benefit of ICI therapy [89]. TCR repertoire can be assessed using high-throughput sequencing at the bulk and single-cell level coupled with computational analysis in tumor and blood samples [25,44,90,91].

In the NEOSTAR trial, the increased T cell richness and clonality in resected tumors from a limited number of early-stage NSCLC patients was more profound in patients treated with nivolumab plus ipilimumab versus nivolumab, suggesting that there is a potential ICI-induced tumor infiltration from peripheral blood that may result from immunologic priming [38].

Forde et al. assessed the frequency of tumor-specific T cell clones in tumor and peripheral blood from neoadjuvant nivolumab in resectable NSCLC [21]. The author observed that patients that achieved MPR after neoadjuvant immunotherapy have a higher frequency of shared T cell clones between the tumor and peripheral blood than patients with no MPR, and that, at the time of surgery, peripheral blood samples had an expansion of T cell clones that were not found in pre-treatment peripheral blood [21,92]. In samples from the same clinical trial, Caushi et al. [93] evaluated mutation-associated neoantigen (MANA)-specific T cell clones in peripheral blood and observed that seven out of the 19 specific T cell responses for MANA that were detected one day before treatment were still detected by day 44, and that new MANA T cell responses developed by day 44. One patient with a MPR T cell clone specific for MANA rapidly expanded in peripheral blood after neoadjuvant nivolumab, and three of those clones were also found in the primary resected tumor and lymph node [93]. A later study evaluated the transcriptomic profile of MANA-specific TILs, and, among other findings, T-cell dysfunction programs were observed in non-MPR MANA-specific T cells, whereas MPR MANA-specific cells have programs associated with memory and effector function [93].

In one exploratory study, the TCR repertoire assessed by NGS in pretreatment and post-treatment tissue and peripheral blood samples from patients of the NADIM trial was used to evaluate its predictive value. Uneven TCR repertoire diversity evaluated in tissue samples associated with cPR, the top 1% of clones predicted better cPR than evenness, and both biomarkers together were better predictors of cPR than PD-L1 and TMB. In this study, the clonal space in the peripheral blood of the top 1% of clones in pretreatment tissue samples was significantly reduced in patients with cPR, and no significant reductions were observed in patients without cPR. Interestingly, comparing the immune gene expression profile of pretreatment samples from patients with a high and low top 1% clonal space, several immune gene signatures such as Interferon gamma and IL2 were differentially expressed, suggesting that this biomarker can be used to predict response to therapy and to evaluate tumor immunogenicity [94].

Large-scale studies are still needed to investigate the T cell repertoire in patients with resectable NSCLC with neoadjuvant immunotherapy [13,92].

### 3.6. Tertiary Lymphoid Structures (TLS) and B-Lymphocytes

TLS are organized lymphoid structures composed of aggregates of B-cells, T-cells, plasma cells, follicular dendritic cells, and high endothelial venules. These aggregates develop in tumor tissue because of the complex interaction of stroma cells, immune cells, and tumor cells. TLS are considered to be centers of the initiation of effective anti-tumor immune response, including the presentation of neoantigens to T cells and dendritic cells and the activation, proliferation, and differentiation of T and B cells [95]. TLS can be observed in tumor tissue at different levels of maturation. Mature TLS are characterized by displaying a germinal center, and they can be assessed by IHC using CD21 and CD23, which highlights immature and mature follicular dendritic cells [90,95]. The presence of TLS has been associated with a better response to ICI therapy in different cancer types including NSCLC [91,95,96,97,98,99,100,101,102]. B cell subpopulations have been described as having different, important roles in tumorigenesis. They can act indirectly as an antigen-presenting cell, enhance effector T cell response by secreting immunostimulatory cytokines such as IL2, IL4, INF-gamma, and TNF-alpha, among others, and differentiate into plasma cells and produce anti-tumor-cell antibodies that can activate, complement, and promote cellular cytotoxicity [103]. A special B cell subpopulation, regulatory B cells, may impair anti-tumor immune response by producing immunosuppressive cytokines such as IL10 and TGF-beta [104].

Histological studies of the tumor bed of resected NSCLC from patients treated with neoadjuvant therapy have highlighted the presence of TLS, defined as lymphoid aggregates with a germinal center and high endothelial venules assessed by H&E staining, as a feature of immune activation in tumor samples with pCR and MPR [35].

### 3.7. Circulating Tumor DNA

Circulating tumor DNA (ctDNA) is a cell-free DNA molecule that is released by apoptotic and necrotic tumor cells into the bloodstream, contains matched somatic mutations, and has been suggested as a sensitive biomarker to assess tumor response after neoadjuvant therapy and correlate clinical benefit [105]. Reduced absent levels of ctDNA may potentially predict the prolonged survival of patients treated with ICIs [106]. Because ctDNA sequencing of the tumor landscape may be captured using blood-based TMB (bTMB) measurement, some studies have evaluated its predictive value in the response to therapy [107,108].

Although there is limited data on the predictive value of ctDNA to assess efficacy of neoadjuvant treatment, the results of the Checkmate 816 and NADIM trials suggest that ctDNA in a pretreatment sample is a potential early predictor of recurrence-free survival, and that this may be a better predictor than radiologic assessment. In a subset of patients, the phase 3 checkmate 816 trial evaluated ctDNA using a tumor-guided panel for WES. ctDNA clearance was defined as a presurgical change from detectable ctDNA before cycle 1 to ctDNA negative before cycle 3. Patients with ctDNA clearance had a higher percentage of cPR compared to patients without ctDNA clearance; patients treated with nivolumab and chemotherapy had a higher percentage of ctDNA clearance compared to patients treated with chemotherapy alone. Finally, ctDNA clearance was also associated with long event-free survival in both treatment arms [21]. Similarly, in the NADIM trial, low ctDNA levels assessed by NGS in baseline blood samples were associated with a better progression-free survival and overall survival, and the assessment of ctDNA predicted survival better than radiological assessment [12,109].

### 3.8. Circulating Peripheral Immune Cell Subsets and Cytokines

In NSCLC patients, there are dynamic changes in peripheral immune cell populations and soluble proteins, including cytokines, following neoadjuvant immunotherapy [13]. Methods to identify immunophenotypes potentially related to ICI treatment efficacy in the peripheral blood include plasma and serum-based cytokine multiplex detection platforms and/or arrays, flow cytometry, and CyTOF [13,110]. An investigation of immune cell subsets and cytokine changes in the peripheral blood has been performed in several clinical trials of neoadjuvant immunotherapy for resectable NSCLC. In the LCMC3 trial, different immune cell subset populations and potential therapy-induced modulation were assessed in pre- and post-treatment blood samples from NSCLC patients who received neoadjuvant atezolizumab. Patients who achieved MPR had a lower density of specific subsets of T-cells and natural killer subsets compared to patients that did not achieve MPR [110]. Similarly, in the NADIM trial, higher levels of soluble proteins associated with immune activation, such as 4-1BB, were observed in patients who achieved a pCR as compared to those that did not. Interestingly, prior to therapy, there was a trend towards a higher expression of PD-1 on CD4+ T cells, CD8+ T cells, and NK cells in circulation in patients who achieved pCR after neoadjuvant therapy, as well as a higher mean fluorescence intensity of cytotoxic markers such as NKG2D and CD56 on CD8+ T cells [111].

### 3.9. Complete Blood Count (CBC)

Complete blood counts based on the absolute values and ratios of circulating blood cells have been suggested as potential predictor markers of tumor response to ICIs [111,112]. Changes in peripheral blood count occur as the tumor stage progress and neoadjuvant ICIs are received. Laza-Briviesca et al. [111], in the NADIM study, reported that, in patients with early-stage NSCLC who received neoadjuvant chemotherapy plus nivolumab, a decrease in the total peripheral of leucocytes, eosinophils, monocytes, neutrophils, erythrocytes, and platelets was observed, as well as discreet changes in lymphocytes, basophils, and lactate dehydrogenase (LDH). In the same study, post-treatment results showed a decrease in peripheral NLR, with the M:L ratio and platelets-to-lymphocytes ratio (PLR) being significantly lower in the blood of patients who achieved pathologic complete response (pCR) [111].

### 3.10. Gut Microbial-Derived Metabolites

Several studies have suggested that bacteria residing in the gut may determine anti-cancer therapy efficacy with ICIs, due to its extensive influence in systemic immune environment [113,114,115]. In the NEOSTAR trial, the gut microbiome was assessed using targeted 16S ribosomal RNA gene sequencing and compared among groups based on MPR status and nivolumab and nivolumab plus ipilimumab arms. No difference in diversity was found based on MPR status. *Paraprevotella* and *Akkermansia* spp. were associated with MPR in both arms, and an unclassified *Ruminococcus* sp. was associated with MPR in the nivolumab plus ipilimumab arm. *Dialister* sp. Was associated with a decrease in toxicity to nivolumab. Bifidobacterium and *Enterobacter* spp. and an unclassified genus of *Erysipelotrochaceae* was associated with a reduced toxicity to dual therapy. *Akkermansia* sp. and *Bifidobacterium*
*were* correlated with TCR clonality, and *Anaerofustis* sp. *Faecalibaculum* sp. And *Ruminococcus_1* sp. were associated with T cell richness [38].

### 3.11. Other Host-Related Biomarkers

Other host immune-related biomarkers may be predictive of the response to neoadjuvant therapy in early-stage NSCLC patients.

Several studies have shown that there are sex-based differences in the immune profile of lung cancer tumors and in the therapy efficacy of ICIs [116,117,118,119,120,121]. Overall, women have a stronger innate and adaptive immune response compared to men [119,120,121]. Additionally, surgical resected lung adenocarcinoma from female patients were found to have a higher infiltration of tumor-associated immune cells compared to male patients [121]. Furthermore, several studies have shown that, overall, the treatment response with ICIs might be sex-dependent. Male patients seemed to have more benefit from ICIs alone versus a control when compared to female patients, and female patients exhibited more clinical benefits from ICIs plus chemotherapy versus a control when compared to male patients [117,119,122,123,124]. In addition, female patients had higher immune-related adverse events [125]. These sex-based differences may be related to sex hormones and sex-related genes, since many genes on the X chromosome regulate diverse aspects of immune response (e.g., Toll-like receptors, cytokine receptors such as IL2RG, and transcription factors such as FOXP3) [126,127,128]. Of note, some biomarkers, such as TMB, may have a greater predictive power in female patients with NSCLC versus in male patients with NSCLC [123,125].

There is still limited data on sex-based differences with regard to clinical benefit in patients treated with neoadjuvant immunotherapy in resectable NSCLC. In the Checkmate 816 trial, the median EFS was longer in female patients in both arms, specifically in patients treated with nivolumab plus chemotherapy. The median EFS was not reached in female patients, and, in male patients, the median EFS was 30.6 months. However, only a slight difference was seen in pathological complete response in female (27.5%) compared to male (22.7%) patients in the Nivolumab plus chemotherapy arm [12]. Larger studies are needed to elucidate the role of sex as a predictive biomarker for the response to immunotherapy in these settings.

Similarly, other factors that are associated with a distinct anti-tumor immune response, such age, body mass index, and Human Leukocyte Antigen-1 (HLA-I) status, may be further studied to better stratify early-stage NSCLC patients for neoadjuvant immunotherapy [129,130,131,132].

## 4. Conclusions and Future Directions

Several clinical trials of neoadjuvant immunotherapy in resectable NSCLC have shown promising results and have led to the recent approval of neoadjuvant immunotherapy in resectable NSCLC. The proposal to use cPR and MPR as a surrogate endpoint of overall survival in clinical trials has accelerated the development of clinical trials, including immunotherapy in neoadjuvant settings. The assessment of PR in resected samples represents an opportunity to study the immune changes in the tumor bed that correlate with response, as well as to identify mechanisms of resistance to therapy. Further studies using multiplex or high-plex techniques and computational tools will allow for a better assessment of the different immune cell subsets, the spatial interactions among them, and the architectural patterns of the tumor microenvironment. It is worth mentioning that correlative studies in the aforementioned trials have allowed for the interrogation of several biomarkers that can be evaluated in tumors, blood, and stool, and they have provided precious information regarding the dynamics of the immune response in patients treated with neoadjuvant immunotherapy. There are several challenges that are now faced with respect to neoadjuvant treatment in resectable NSCLC, including the validation of cPR and MPR as predictors of long-term survival, the use of adjuvant therapy, and the stratification of patients according to genomic alterations as well as to the identification of biomarkers or pathways that are associated with resistance to immunotherapy. Successful designs of prospective clinical trials of neoadjuvant immunotherapy will require a multidisciplinary team of clinicians and investigators, as well as the use of immune monitoring strategies that can integrate the analysis of histology-related features, genomic, transcriptomic, and phenotypic profiling, and functional assays.

## Figures and Tables

**Figure 1 cancers-14-02775-f001:**
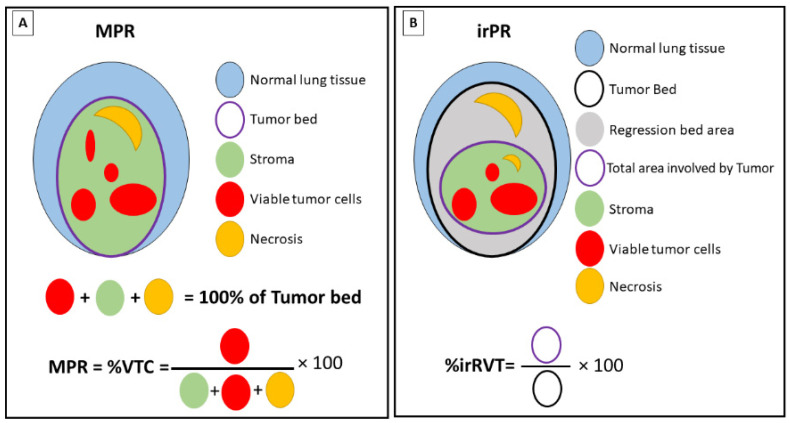
Histologic assessment of major pathologic response and immune-related pathologic complete response. (**A**) Schematic representation of the tissue components for the assessment of major pathologic response (MPR) in sample specimen of primary tumor. The tumor bed is constituted by viable tumor cells (VTC), stroma (including inflammatory cells and fibrosis), and necrosis. All the components in the tumor bed area have a sum of 100% (**B**) Schematic representation of the tissue components for the assessment of the immune-related pathologic response (irPR) in sample specimen of primary tumor. The tumor bed contains the regression bed, where immune-related histologic features can be observed, and an inner area involved by tumor that is constituted by viable tumor cells, stroma (including inflammatory cells and fibrosis), and necrosis. The percentage of residual viable tumor (irRVT) is assessed by dividing the total surface area of RVT (circled in purple) by the total tumor bed area (circled in black) ×100.

**Figure 2 cancers-14-02775-f002:**
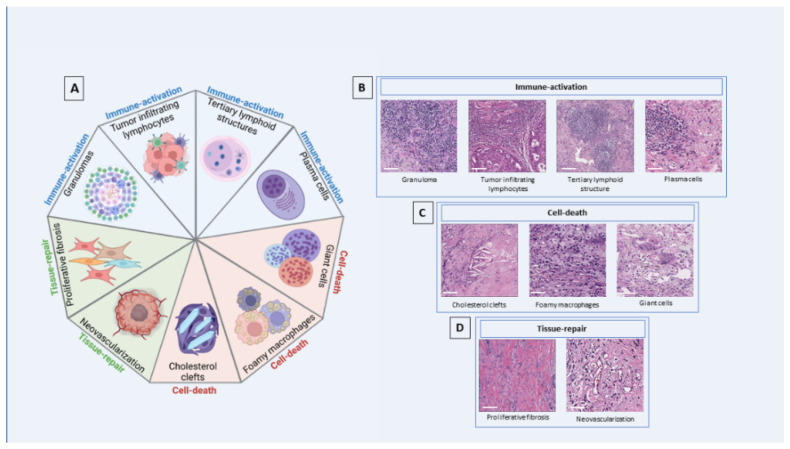
Histopathologic characteristics of pathologic response after neoadjuvant immune checkpoint inhibitors. (**A**). Components of immune-mediated tumor regression. Histologic features grouped with regard to phenomena of immune activation (**B**), cell death (**C**), and tissue repair (**D**). Scale bar: 100 μm. Graphic created in part using Biorender (http://biorender.com, access date 24 April 2022).

**Figure 3 cancers-14-02775-f003:**
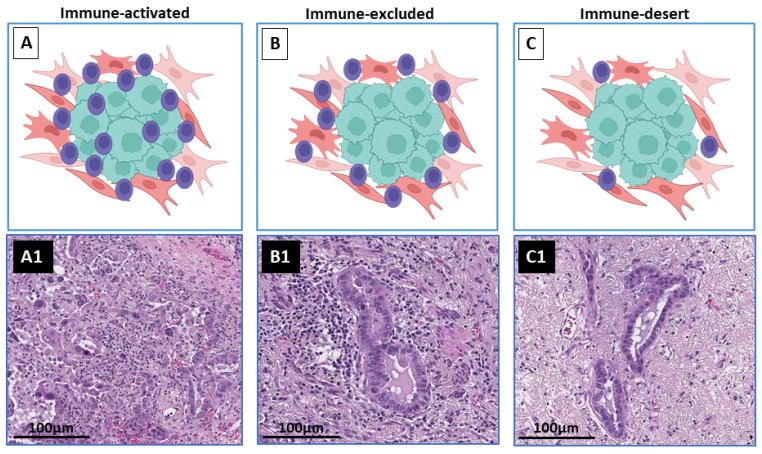
Tumor-immune responsiveness profile. Schematic representation of tumor response to immune checkpoint inhibitors. (**A**) Immune-activated, characterized by high degree of tumor-inflammatory infiltrate; (**A1**) Representative H&E image showing immune-activated tumor immune profile. (**B**) Immune-excluded, characterized by presence of inflammatory cells in the tumor nest margin with no compromise of the tumor cells; (**B1**) Representative H&E image showing immune-excluded tumor immune profile. (**C**) Immune-desert, characterized by absence of inflammatory cells within tumor nest and tumor margin; (**C1**) Representative H&E image showing immune-desert tumor immune profile. Graphic created in part using Biorender (http://biorender.com, access date 27 April 2022).

**Figure 4 cancers-14-02775-f004:**
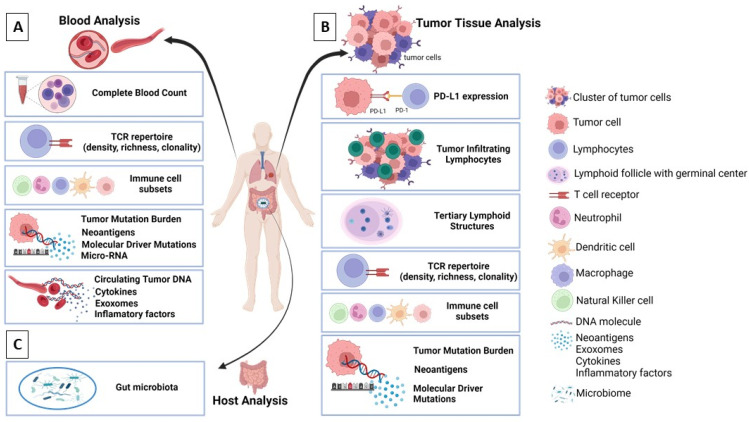
Biomarkers used for assessment of response to neoadjuvant immunotherapy and patient monitoring in resectable NSCLC. Biomarkers grouped by source: (**A**) blood, (**B**) tumor tissue, and (**C**) stool. PD-L1, programmed cell death-ligand 1; PD-1, programmed cell death-1; TCR, T cell receptor. Graphic created using Biorender (http://biorender.com, access date 27 April 2022).

**Table 1 cancers-14-02775-t001:** Representative clinical trials for neoadjuvant immunotherapy alone or in combination with chemotherapy in resectable non-small-cell lung cancer.

Trial Name (Registry Number) Phase	Tumor Stage	Patient *N*	Neoadjuvant Treatment	MPR	pCR	Outcome	PD-L1 (IHC)	Correlative Studies
>Checkmate 159 (NCT02259621) Phase 2	IB–IIIA	45	Arm A: Nivolumab Arm B: Nivolumab + Carboplatin + Paclitaxel	45%	22%	30-months disease-free: 15/20 patients Median RFS: not reached-24-months RFS: 69%	Yes % PD-L1+(≥1%): 46.6% (7/15)	Tumor mutation burden Molecular mutations Circulating tumor DNA Tumor infiltrating lymphocytes TCR repertoire
NEOSTAR (NCT03158129) Phase 2	IA–IIIA	88	Arm A: Nivolumab Arm B: Nivolumab + Ipilimumab Arm C: Nivolumab + Platinum doublet CT Arm D: Nivolumab + Ipilimumab + Platinum doublet CT	Arm A: 22% Arm B: 38% Arm C and D: not reported	Arm A: 9% Arm B: 29% Arm C and D: not reported	Median OS and Lung cancer-related RFS: not reached after a median follow-up of 22.2 months	Pretherapy tumor PD-L1: MPR: median, 3% No MPR: median, 0% Posttherapy tumor PD-L1: MPR: median, 5% No MPR: median, 0%	Flow cytometry Multiplex immunofluorescence T-cell receptor sequencing Gut microbiome
LCMC3 (NCT02927301) Phase 2	IB–IIIB (resectable)	179	Atezolizumab	20%	7%	Not reported	Yes PD-L1+ (≥1%): 19.5% (35/179)	Multiplex immunofluorescence Tumor mutation burden Molecular mutations RNA sequencing Flow cytometry
NADIM (NCT03081689) Phase 2	IIIA	46	Nivolumab + Carboplatin + Paclitaxel	83%	71%	PFS (24 months): 7% OS (12 months):97.8% OS (18 months):93.5% OS (24 months): 89.9%	Yes PD-L1 + (≥1%)): 39% (18/46)	Multiplex Immunofluorescence T-cell receptor sequencing Tumor mutation burden Molecular mutations Circulating tumor DNA
MK3475-223 (NCT02938624) Phase 1	I-II	28	Pembrolizumab	40%	Not reported	Not reported	Yes PD-L1 + (≥1%): 18% (5/28)	Not reported
NEOCOAST (NCT03794544) Phase 2	I–IIIA (resectable)	160	Arm A: Durvalumab Arm B: Durvalumab + Oleclumab Arm C: Durvalumab + Monalizumab Arm D: Durvalumab + Danvatirsen	Not reported	Not reported	Not reported	Yes, not reported	Tumor genomics Tumor microenvironment and T cell population
PRINCEPS (NCT02994576) Phase 2	resectable	60	Atezolizumab	14%	Not observed	Not reported	Yes, not reported	Multiplex Immunofluorescence Molecular mutations SUVmax
SAKK (NCT02572843) Phase 2	IIIA (resectable)	67	Durvalumab + CT	62%	18%	1-year EFS: 73% Median EFS and OS: not reached after 28.6 months follow-up.	Yes, not reported	Not reported
Checkmate 816 (NCT02998528) Phase 3	IB–IIIA	358	Platinum doublet CT Nivolumab + platinum doublet CT	Platinum doublet CT: 8.9% Nivolumab + platinum doublet CT: 36.9%	Platinum doublet CT: 2.2% Nivolumab + platinum doublet CT: 24.0%	Platinum doublet CT: Median EFS: 20.8 months 1-year OS: 63.4% 2-year OS:45.3% Nivolumab + platinum doublet CT: Median EFS: 31.6 months 1-year OS: 76.1% 2-year OS: 63.8%	Yes PD-L1 + (≥1%): 49% (178/358)	Tumor mutation burden Circulating Tumor DNA
Impower 030 (NCT03456063) Phase 3	II–IIIA-selected IIIB	451	Arm A: Atezolizumab + platinum doublet CT Arm B: Placebo + platinum doublet CT	ongoing, end date April 2024	ongoing, end date April 2024	Not reported	Yes, not reported	Not reported
AEGEAN (NCT03800134) Phase 3	II–III	824	Arm 1: Durvalumab + platinum doublet CT Arm 2: placebo + platinum doublet CT	ongoing, end date April 2024	ongoing, end date April 2024	Not reported	Yes, not reported	Not reported

Abbreviations: N, number; MPR, major pathological response; cPR, complete pathological response; CT, chemotherapy; EFS, event-free survival; RFS, recurrence-free survival, OS, overall survival; PFS, progression-free survival.

**Table 2 cancers-14-02775-t002:** Assays used to assess biomarkers in tissue, blood, and gut micriobiome.

Biomarker	Source	Gold Standard	In Development
PD-L1 expression	Tissue	Immunohistochemistry	Multiplex immunoflourescence
Tumor-infiltrating lymphocytes (TILs)	Tissue	H&E stain: Pathology analysis	Immunohistochemistry Multiplex immunofluorescence/High-plex technologies Next Generation Sequencing Flow Cytometry/CyTOF TCR and BCR sequencing
Tertiary lymphoid structures (TLSs)	Tissue	H&E stain: Pathology analysis	Immunohistochemistry Multiplex immunofluorescence/High-plex technologies Next Generation Sequencing
Immune cell subsets	Tissue	Immunohistochemistry	Multiplex immunofluorescence/High-plex technologies Next Generation Sequencing Flow Cytometry TCR and BCR sequencing
Circulating immune cell subsets	Blood	Flow Cytometry	Functional T cells assays ELISPOT Next Generation Sequencing Cytokines/chemokines CyTOF
T cell receptor repertoire (TCR)	Tissue, Blood	None	TCR and BCR sequencing
Tumor mutation burden (TMB)	Tissue, Blood	Whole exome sequencing (WES)	Next Generation Sequencing
Complete Blood Count (CBC)	Blood	Hemogram	
Circulating tumor DNA (ctDNA)	Blood	None	Next Generation Sequencing
Gut microbiota	Stool	None	Next Generation Sequencing

Abbreviations: T cell receptor (TCR); B cell receptor (BCR); CyTOF, mass cytometry.

## Data Availability

No new data were created or analyzed in this study. Data sharing is not applicable to this article.

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
