# Peer review of "Pathological Response and Immune Biomarker Assessment in Non-Small-Cell Lung Carcinoma Receiving Neoadjuvant Immune Checkpoint Inhibitors"

_cancers, 2022, doi:10.3390/cancers14112775_

Round 1

Reviewer 1 Report

I fully agree with the authors that the assessment of immune changes differs in surgically resected specimens from patients treated with immunotherapy. Therefore, the study of biomarkers that can predict the response to therapy and control the response to treatment is certainly relevant. In this review, the authors provided information on current guidelines for assessing pathological responses in surgically resected NSCLC tumors treated with neoadjuvant immunotherapy and described potential biomarkers that have been used to predict the benefit of neoadjuvant immunotherapy in patients with resectable NSCLC.

I would like to see the methodology for preparing the review, on what basis the selection of publications for the preparation of the review was carried out, which keywords were used for the search, for which databases, etc.

Author Response

Point 1: I would like to see the methodology for preparing the review, on what basis the selection of publications for the preparation of the review was carried out, which keywords were used for the search, for which databases, etc.

Response 1: We thanks to the reviewer comments, and herein we provide information on the basis for the selection of publications for this review. We performed the literature review about this research topic using Pubmed, Google Scholar, and Google search engine using the following keywords: neoadjuvant + immunotherapy + NSCLC; neoadjuvant therapy + resectable NSCLC; pathological response or major pathological response in NSCLC; irPR + NSCLC; Biomarkers + resectable NSCLC. We selected peer review articles related to neoadjuvant therapy in NSCLC including immunotherapy. We also selected abstracts presented in national and international scientific meetings that provided preliminary data from clinical trials of neoadjuvant immunotherapy. 

Reviewer 2 Report

Dear authors,

Your review provide a comprehensive overview of the pathological response and immune biomarker assessment in non-small cell lung cancer patients receiving neoadjuvant immune checkpoint inhibitors. The manuscript is well written, and I believe it could be of interest for the scientific community. I have some comments / suggestion:

  1. Line 57: Please check the references.
  2. Line 70: PR acronym not previously reported (check the abstract as well where PR acronym was not previously reported – line 24)
  3. Line 138: Please specify which standard chemotherapy seems to activate the immune response.
  4. As for Table 1:
    1. As stated by the authors, cPR and MPR are surrogate predictors of survival. Did the studies that you mentioned reported any information on the Overall survival (median, or at 2, 5 years) for these patients as well? If so, I suggest the authors to provide this information it in the table or in the text.
    2. Please provide the number of patients included in the clinical trials that you mentioned.
    3. What was the mean of PD-L1 expression in the included studies? Is it reported?
    4. What do you mean with representative? Did you include all clinical trials about neoadjuvant ICI in patients with resectable NSCLC? How did you select representative studies?
  5. In Table 1 you reported clinical trials, but there is any evidence coming from the real-word setting?
  6. Line 157: Please check punctuation
  7. Line 166: Is this study included in Table 1? If so, please reported also NCT number.
  8. Figure 1: VTC acronym not reported
  9. Line 213: Please check the text color
  10. Line 270-275: I find this is an interesting point and that a machine learning based approach could be useful to evaluate PR. Did the authors that you mentioned report anything about validity measures (e.g., sensitivity, positive predictive value) of such approach?
  11. The authors reported a comprehensive overview of the immune biomarkers, and pathological response of neoadjuvant ICIs. However, I believe that also another old prognostic variable such as sex should been taken into consideration. Several studies demonstrated that male and female patients have a different immune profile (pmid: 31487832) that could affect response to immunotherapy, and also observational studies reported a different survival of male and female patients with NSCLC on the basis of histology (pmid: 34885238). I suggest authors to report this missing point in the text of your manuscript.

Author Response

Dear authors,

Your review provide a comprehensive overview of the pathological response and immune biomarker assessment in non-small cell lung cancer patients receiving neoadjuvant immune checkpoint inhibitors. The manuscript is well written, and I believe it could be of interest for the scientific community. I have some comments / suggestion:

Point 1: Line 57: Please check the references.

Response 1: We thanks to the reviewer for highlighting this typo. We have reviewed the references for that phrase as suggested and we corrected the misplaced references in the corrected version of the manuscript. Thank you.

Point 2: Line 70: PR acronym not previously reported (check the abstract as well where PR acronym was not previously reported – line 24)

Response 2: We thanks to the reviewer for pointing this out. We have reviewed and added the definition of the acronym as first cited in Line 25 of the abstract section and Line 69 of the introduction section.

Point 3: Line 138: Please specify which standard chemotherapy seems to activate the immune response.

Response 3: Per the reviewer suggestion, we have included the information of the chemotherapy agents (cisplatin combined with gemcitabine or docetaxel, and carboplatin combined with paclitaxel or pemetrexed) included in the study cited which describes activation of immune response after neoadjuvant chemotherapy. The changes are included in the following text:

Lines 139-143: “It has also been suggested that some chemotherapy agents including cisplatin combined with gemcitabine or docetaxel, and carboplatin combined with paclitaxel or pemetrexed, activates the immune response of NSCLC patients, since PD-L1 and specific T-cells immune cells are higher in NSCLC resected tumors that received neoadjuvant chemotherapy compared to chemonaïve-tumors [36].”

Point 4: As for Table 1:

As stated by the authors, cPR and MPR are surrogate predictors of survival.

Did the studies that you mentioned reported any information on the Overall survival (median, or at 2, 5 years) for these patients as well? If so, I suggest the authors to provide this information it in the table or in the text.

Response 4: Thank you for the reviewer suggestion. We agree that adding that information is important. We have added new columns in Table 1 including among other information, Overall survival and other relevant outcomes reported on the clinical trials. Please, check the new version of the table 1. Line 166.

Point 5: Please provide the number of patients included in the clinical trials that you mentioned.

Response 5: number of patients included in the clinical trials has been added to the new version of the table 1. Please, check line 166.

Point 6: What was the mean of PD-L1 expression in the included studies? Is it reported?

Response 6: We thanks the reviewer’s suggestion, for those clinical trials that reported PD-L1 assessed by immunohistochemistry, the information is now included in the new version of the manuscript, Table 1. Please, check line 166.

Point 7: What do you mean with representative? Did you include all clinical trials about neoadjuvant ICI in patients with resectable NSCLC? How did you select representative studies?

Response 7: In this review we only included information of clinical trials with availability of preliminary data of pathology response and/or correlative studies.

Point 8: In Table 1 you reported clinical trials, but there is any evidence coming from the real-word setting?

Response 8: Dear reviewer we appreciate this comment. There is still no data available from the real world setting since this indication has just being approved by FDA (03/04/2022) for NSCLC treatment. Therefore, the compiled information only refers from the clinical trials settings.

Reference: https://www.fda.gov/drugs/resources-information-approved-drugs/fda-approves-neoadjuvant-nivolumab-and-platinum-doublet-chemotherapy-early-stage-non-small-cell-lung

Point 9: Line 157: Please check punctuation

Response 9: Dear reviewer we added the comma before “Table 1” in Line 160 of the current version..

Point 10: Line 166: Is this study included in Table 1? If so, please reported also NCT number.

Response 10: The information from that clinical trial is included in the main manuscript, we thanks the reviewer for pointing out this omission, and we have included this clinical trial in table 1. Line 166 of the revised manuscript.

Reference: https://www.nejm.org/doi/full/10.1056/nejmoa1716078

Point 11: Figure 1: VTC acronym not reported

Response 11: VTC stands for viable tumor cells. We have now corrected this oversight and included the description of the acronym in the figure legend. Line 189.

Point 12: Line 213: Please check the text color

Response 12: Red dot has been changed. Line: 217.

Point 13: Line 270-275: I find this is an interesting point and that a machine learning based approach could be useful to evaluate PR. Did the authors that you mentioned report anything about validity measures (e.g., sensitivity, positive predictive value) of such approach?

Response 13: Dear reviewer thanks for highlighting this topic. The authors did not mention validity measures and the results has not still been published. We have clarified this point in the following text:

Line 275-279: “Dacic et al quantified and measured tumor bed area and identified residual viable tumor cells showing strong correlation between AI tool and manual MPR assessment although with different percentage ranges probably due to differences in methodology, however, no sensitive or positive predictive value has been presented [56].”

Point 14: The authors reported a comprehensive overview of the immune biomarkers, and pathological response of neoadjuvant ICIs. However, I believe that also another old prognostic variable such as sex should been taken into consideration. Several studies demonstrated that male and female patients have a different immune profile (pmid: 31487832) that could affect response to immunotherapy, and also observational studies reported a different survival of male and female patients with NSCLC on the basis of histology (pmid: 34885238). I suggest authors to report this missing point in the text of your manuscript.

Response 14: We thank the reviewers input about this important topic. We have included a new section on our review regarding “Other host-related biomarkers: and we included information about patient’s sex and immune response. Please see line: 597 of the revised manuscript.
